# Synergic Effects of Nano Additives on Mechanical Performance and Microstructure of Lightweight Cement Mortar

Yiying Du and Aleksandrs Korjakins *

Department of Building Materials and Products, Faculty of Civil Engineering, Riga Technical University, Kipsalas St. 6A, LV-1048 Riga, Latvia; yiying.du@edu.rtu.lv
* Correspondence: aleksandrs.korjakins@rtu.lv; Tel.: +371-26422442

**Abstract:** Owing to their convenient manufacture, transportation, low energy consumption, and environmental impacts, lightweight cement composites have been applied as building and construction materials. However, its decreased density is associated with a reduction in mechanical strength. In most existing investigations, attempts have been made to improve mechanical behaviours via supplementary cementitious or fibre materials, whereas limited studies have been implemented on the effects of nano additives, especially their synergic influence. In this study, industrial waste fly ash cenosphere (FAC) has been utilized as lightweight aggregate by 73.3% cement weight to fabricate sustainable lightweight cement mortar (LWCM). Carbon nanotubes (CNTs) at a dosage of 0.05%, 0.15%, and 0.45% and nano silica (NS) with the content of 0.2%, 0.6%, and 1.0% by cement weight have been applied as modifying additives. Experiments were carried out to test flexural strength, compressive strength, and water absorption. SEM, TG, and XRD analyses were conducted to evaluate microstructure and hydration characteristics. Based on the outcomes, the inclusion of CNTs and NS can effectively increase flexural and compressive strength and reduce absorbed water weight. The analysis of SEM, TG, and XRD reveals that the binary usage of CNTs and NS can improve pore structure and facilitate hydration reaction.

**Keywords:** carbon nanotubes; compressive strength; fly ash cenospheres; flexural strength; hydration; microstructure; nano Silica; water absorption

## 1. Introduction

Lightweight concrete (LWC) and lightweight cement composites (LWCC) have been widely regarded as promising materials for the building and construction industry for their superior advantages in engineering properties such as low thermal expansion, high fire resistance, and outstanding heat and sound insulation [1]. The decreased weight contributes to convenient fabrication, shipping, transportation, and installation, and leads to a significant reduction in material consumption, energy requirements, and construction costs [2,3]. Based on unit weight, concrete materials can be classified into three categories: (a) heavyweight concrete, with a density of 3200 to 4000 kg/m$^3$; (b) normal concrete, with a unit weight from 2400 to 2600 kg/m$^3$; and (c) LWC, with a density less than 1920 kg/m$^3$ [4]. Traditionally, LWC and LWCC are produced via the addition of lightweight fillers as aggregates into the cement [4]. These fillers alter the mechanical properties and durability of the material, and their effects depend on the characteristics of the fillers as well as the bonding effect between aggregates and cement matrix [5]. Lightweight aggregates can be classified into (a) natural aggregates such as pumice, (b) organic aggregates like palm oil shells and rubber seeds, and (c) synthetic aggregates such as waste glass and expanded polystyrene beads [6–10].

Fly ash cenosphere (FAC) is also a widely incorporated artificial lightweight aggregate in the fabrication of LWC and LWCC due to its special physicochemical characteristics and outstanding engineering performance, such as low density and superior thermal

insulation [11]. The properties of FAC rely on a variety of parameters such as the category of the coal, grinding approaches, combustion procedures, and withdrawal process [12]. As with fly ash, FAC is an industrial waste yielded from thermal power plants as a by-product of the combustion of coal. According to Dixit et al. [13], the production of concrete results in a tremendous consumption of aggregates, with a demand in the building sector of approximately 49% raw stones and sand, 25% wood, and 16% water. This contributes to great concerns about their future availability and the environmental effects of their manufacture. Under these circumstances, the introduction of recycled aggregates, especially the reuse of industrial wastes, is regarded as a solution to the raw material shortage and meanwhile can improve the utilization rate of materials by converting wastes into resources [14]. In addition, the fabrication of cement is associated with considerable environmental burdens, such as climate change and global warming, and requires intensive energy. Around 7% of global anthropogenic emissions of carbon dioxide, as well as 14% of the world's industrial energy consumption, are attributed to the production of cement [15]. Thus, the incorporation of FAC in the manufacture of lightweight building and construction materials is a sustainable movement for environmental conservation as well as the preservation of natural resources.

The presence of FAC can remarkably reduce the density of cement-based materials, which is ascribed to the incorporation of a great number of small air voids, for FAC is difficult to pack densely due to its spherical shape [11]. However, along with reduced density, a decrease in compressive, flexural, and tensile strength is observed in lightweight cement materials with FAC, similar to other lightweight aggregates. Based on the analysis by Danish et al. [11], a strong linear evolution was witnessed between the compressive strength of LWC and the density of FAC. Along with the growth of the percentage of FAC, the compressive strength decreased in a linear pattern. Efforts have been made in order to ameliorate these mechanical behaviors and develop high-strength LWCC, by introducing supplementary cementitious materials. In this regard, for instance, Wu et al. [16] combined silica fume into LWCC with FAC and succeeded in fabricating LWCC with compressive strength up to 69.4 MPa and flexural strength to 7.3 MPa. Satpathy et al. [1] carried out experiments on LWCC containing both FAC and sintered fly ash, proposing that a composite with 50% FAC and 75% sintered fly ash met the requirements of M25 concrete, and a composite with 75% FAC and 50% sintered fly ash can be used as M20 grade concrete. Other attempts have also been made to enhance the brittle nature of lightweight cement-based materials, especially with fibre reinforcement to improve flexural strength and ductility. In this regard, Huang et al. [17] introduced polyvinyl alcohol fibres into the fabrication of LWCC with FAC, iron ore tailings, and fly ash, concluding that FAC was the most effective material to reduce the density of cement composites and improve tensile ductility. Wang et al. [18] compared the reinforcing effects of polyethene and steel fibres on the enhancement of flexural properties of LWCC containing FAC and silica fume. They addressed the fact that the development of flexural strength relied on the category of fibres as well as their bond with cement matrix, and steel fibres exhibited more significant effects than polyethene fibres.

Due to the progress achieved in the field of nanotechnology, the introduction of nanomaterials with particle size finer than 100 nm in producing cement-based materials is identified as sustainable and effective in ameliorating their performance [19]. There is a wide range of nanomaterials used in manufacturing cement-based materials, and among them, nano silica (NS), as well as carbon nanotubes (CNTs), are the two most welcomed in research regarding nano-modification and nano-engineering of cement materials, due to their unique characteristics. The presence of CNTs can act as fibres and effectively bridge the micro-sized and nano-sized gaps in the cementitious matrix, preventing the propagation of micro cracks and improving pore structure [20,21]. The filling effects of CNTs can significantly reduce the porosity and total volume of the cement matrix by filling up micro gaps and pores [22]. Further, CNTs can perform as nucleating sites for hydration reactions, which efficiently facilitate the formation of hydration products [23]. Similar to CNTs, the

nucleation and filler effects of NS, due to its small particle size and great surface area, can contribute to the formation of a denser microstructure of cement materials [24]. In addition, NS can participate in pozzolanic reactions with calcium hydroxide (CH) on the silica surface, producing more calcium silicate hydrate (CSH) gel that fills up the tiny pores in the cement mixture [25]. Most of the studies on NS and CNTs mainly concentrate on their single effects on cement-based materials [21,22,26–29]. Although there have been some attempts made to disclose the synergic influence of NS or CNTs combined with other modifiers such as glass powder, carbon fibres, and fly ash [30–35], very few efforts have been made regarding the hybrid effects of CNTs combined with NS on cementitious materials. Only Sikora et al. [36] implemented compressive and calorimetry experiments on cement pastes reinforced by CNT/NS core-shell additives, showing that the incorporation of an NS shell improved the binding capability between cement matrices and CNTs. Karakouzian et al. [37] studied the synergic influence of CNTs and NS on the compressive, flexural strength and microstructure of cement pastes. They asserted that the addition of high percentages of NS can account for further improvement of mechanical properties of CNTs-cement composites. Narasimman et al. [38] researched compressive behavior, water absorption, and microstructure of LWC containing lightweight expanded clay aggregates under the hybrid effect of CNTs and NS. They found that the incorporation of 1% NS and 2% CNTs led to the peak value of compressive strength after 28-day curing. Garg et al. [39] developed hybrid fibre-reinforced cement composites utilizing a combination of carbon fibres as well as CNTs with micro-silica and NS to improve the dispersion of CNTs. They showed that compared to micro-silica, NS exhibited less effectiveness in promoting the modification effects of CNTs in cement composites. Aydin et al. [40] incorporated NS, CNTs, and fly ash into self-compacting concrete. They showed that the presence of NS worsened the fresh properties of concrete and CNTs contributed to more ductile concrete.

The progress of nanotechnology in the field of building materials also shows the potential to integrate nano additives into lightweight cement-based materials. In this regard, however, limited literature has been published. Only Xi et al. [41] incorporated NS lightweight cement mortar (LWCM) containing FAC, and examined the compressive strength, tensile stress, strain capacity, as well as microstructure. They showed that the addition of NS effectively enhanced the physical characteristics of LWCM. NS showed the potential to restrain the porosity, refine the pore structure, and ameliorate the interfacial transition zone between FAC and cement. Hanif et al. [42] investigated the mechanical properties and microstructure of cement pastes with FAC and NS, pointing out that the presence of NS enhanced the compressive strength of cement pastes owing to the increased interface, the formation of better bonding between FAC particles in the cement matrix, as well as the improved pozzolanic activity of FAC. Najeeb and Mosaberpanah [43] implemented tests on the mechanical performance, durability, and microstructure of LWCM with the inclusion of FAC and NS. The results demonstrated that due to the presence of NS, LWCM showed excellent resistance to high temperatures and sulfuric acid intrusion. The studies concerning the synergic influence of NS-CNTs additives on lightweight cement materials with FAC are not available in the existing research.

In this study, therefore, following sustainable development in the building and construction industries, a green LWCM was fabricated via the utilization of industrial waste FAC as fine aggregates and artificial lightweight aggregates. The type of LWCM belongs to the traditional lightweight aggregate cement mortar. CNTs and NS were added in various contents as synergic modifiers. The objective of the investigation was to discover the binary effects of CNTs and NS on the mechanical performance and microstructure of LWCM. Experiments including a flexural strength test, compressive strength test, water absorption, scanning electron microscopy (SEM), thermogravimetric analysis (TG), and X-ray diffraction (XRD) were carried out to highlight the evolution of mechanical properties as well as the physical and chemical transformation by the changes in morphology and hydration process.

## 2. Materials and Methods

### 2.1. Materials

The cement used in this study was ordinary Portland cement (OPC) produced by Schwenk, conforming to EN 197-1: 2011. Table 1 illustrates its material characteristics. The type of OPC used was CEM 1 42.5 N with an average bulk density of 0.9 to 1.5 g/cm$^3$, ignition loss of 5.0%, early referencing compressive strength after 2 days over 10.0 MPa, and 28-day standard strength varying from 42.5 to 62.5 MPa. Normal tap water was utilized for the preparation of LWCM samples. FAC of Grade 1 as a substitute for ordinary river sand was adopted as fine aggregates to fabricate LWCM. As is shown in Table 2, the color of FAC varies from grey to white, and the shape is a hollow sphere. The bulk density is 0.37 to 0.40 g/cc, and the pH value is 7–8. The morphology of FAC under the microscope is exhibited in Figure 1. As can be seen from the micrograph, the size of FAC varies from 30 to 300 μm and the particles are of both smooth and rough surfaces.

**Table 1.** Material characteristics of OPC used in the experiments.

| Material Property | Values |
|---|---|
| Initial setting time/min | >60 |
| Loss on ignition/% | <5.0 |
| Bulk density/(g/cm$^3$) | 0.9–1.5 |
| Sulphate content/% | <3.5 |
| Chloride content/% | <0.1 |
| Early compressive strength (2 days)/MPa | >10.0 |
| Standard compressive strength (28 days)/MPa | 42.5–62.5 |

**Table 2.** Material characteristics of FAC used in the experiments.

| Material Property | Values |
|---|---|
| Size/micron | 40–300 |
| pH value | 7–8 |
| Bulk density/(g/cm$^3$) | 0.37–0.40 |
| True density/(g/cm$^3$) | 0.6–0.7 |
| Thermal conductivity/(w/mk) | 0.036–0.060 |
| Compressive strength/MPa | 20–40 |

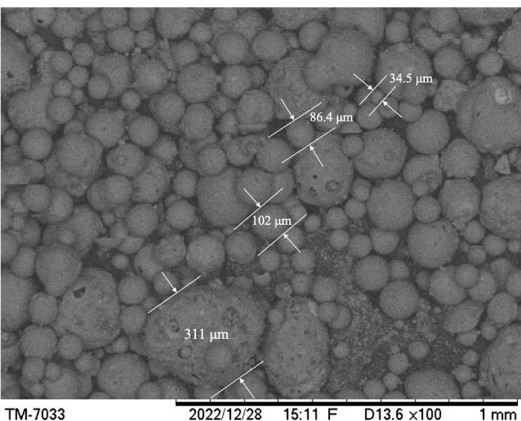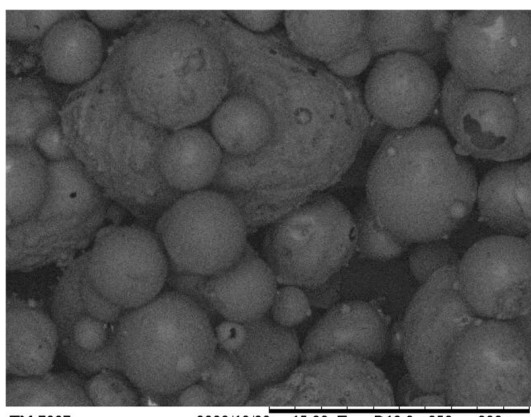

**Figure 1.** Microstructure of FAC.

CNT and NS in the form of powder were used in the investigation to modify the mechanical properties of LWCM. Multi-walled CNTs were selected over single-walled CNTs, for they are of lower cost and have less tendency of agglomeration, promoting a better-quality suspension [20]. CNTs were synthesized via the chemical vapour deposition method which is a most commonly used approach for both laboratory and factory.

The specific area of CNTs was 233 $m^2/g$, and the diameters as well as lengths varied from 6–15 nm and 4–13 nm respectively. In order to obtain preferable workability, as well as to facilitate the homogeneous dispersion of nanofillers, Melment F 10 manufactured by BASF Corporation was used as superplasticizer at an amount of 0.5% by cement weight (bcw). It is a powdery sulfonated polycondensation product based on melamine and is usually used as superplasticizer for cement and calcium-sulphate-based materials. The color of the powder is white, and the bulk density is 500 to 800 $kg/cm^3$.

### 2.2. Mix Design

Mix design is demonstrated in Table 3 and the quantity of each raw material is identified. 12 batches of specimens plus one reference sample with no additives were investigated and parameters encompassing the dosage of CNTs as well as NS were studied. 73% of FAC bcw was used as fine aggregate in the experiment. 56% w/c ratio was considered based on the relevant literature to ensure the dispersion quality of nano additives and the workability of cement mortar. According to Manzur and Yazdani [44], in a system with a high w/c ratio, more water was provided for the better dispersion of CNTs. Further, owing to the hollow and spherical shape of FAC, air can be incorporated during the preparation process of the mortar, leading to a higher required amount of water to guarantee consistency and homogeneous flow [45]. The dosage of CNTs and NS is a prerequisite influencing the extent of improvement in the properties of cement materials. Nazari and Riahi [46] mentioned that the growing content of NS can lead to the improvement of the mechanical properties before the content reached the value of 4% bcw. Li et al. [47] and Ji [48] proved that 1% of NS bcw accounted for the optimal mechanical performance of cementitious composites. Mohsen et al. [49], Li et al. [50], and Li et al. [51] indicated that before the content of CNTs exceeded 0.5% bcw, along with the increase in the amount of CNTs, the mechanical behavior was positively enhanced. Based on the literature, the NS dosage of 0.20%, 0.60%, and 1.00%, as well as the CNT content of 0.05%, 0.15%, and 0.45% bcw were selected for the mix design. In Table 3, the abbreviation RS indicates the reference sample with only OPC, FAC, and water; C means the content of CNTs, and S illustrates that of NS. For instance, the index C5S2 indicates the samples containing 0.05% of CNTs and 0.2% of NS. The percentage of the raw materials is based on cement mass.

**Table 3.** Mix Design for the preparation of per cubic meter LWCM.

| Sample | OPC kg/m³ | Water kg/m³ | Water % | Fine Aggregate FAC kg/m³ | Fine Aggregate FAC % | Melment F10 kg/m³ | Melment F10 % | CNTs kg/m³ | CNTs % | NS kg/m³ | NS % |
|---|---|---|---|---|---|---|---|---|---|---|---|
| RS | 527.34 | 295.31 | 56 | 386.72 | 73.33 | - | - | - | - | - | - |
| C5S0 | 527.34 | 295.31 | 56 | 386.72 | 73.33 | 2.66 | 0.50 | 0.27 | 0.05 | - | - |
| C15S0 | 527.34 | 295.31 | 56 | 386.72 | 73.33 | 2.66 | 0.50 | 0.78 | 0.15 | - | - |
| C45S0 | 527.34 | 295.31 | 56 | 386.72 | 73.33 | 2.66 | 0.50 | 2.38 | 0.45 | - | - |
| C5S2 | 527.34 | 295.31 | 56 | 386.72 | 73.33 | 2.66 | 0.50 | 0.27 | 0.05 | 1.05 | 0.20 |
| C15S2 | 527.34 | 295.31 | 56 | 386.72 | 73.33 | 2.66 | 0.50 | 0.78 | 0.15 | 1.05 | 0.20 |
| C45S2 | 527.34 | 295.31 | 56 | 386.72 | 73.33 | 2.66 | 0.50 | 2.38 | 0.45 | 1.05 | 0.20 |
| C5S6 | 527.34 | 295.31 | 56 | 386.72 | 73.33 | 2.66 | 0.50 | 0.27 | 0.05 | | 0.60 |
| C15S6 | 527.34 | 295.31 | 56 | 386.72 | 73.33 | 2.66 | 0.50 | 0.78 | 0.15 | | 0.60 |
| C45S6 | 527.34 | 295.31 | 56 | 386.72 | 73.33 | 2.66 | 0.50 | 2.38 | 0.45 | | 0.60 |
| C5S10 | 527.34 | 295.31 | 56 | 386.72 | 73.33 | 2.66 | 0.50 | 0.27 | 0.05 | 5.27 | 1.00 |
| C15S10 | 527.34 | 295.31 | 56 | 386.72 | 73.33 | 2.66 | 0.50 | 0.78 | 0.15 | 5.27 | 1.00 |
| C45S10 | 527.34 | 295.31 | 56 | 386.72 | 73.33 | 2.66 | 0.50 | 2.38 | 0.45 | 5.27 | 1.00 |

### 2.3. Dispersion of Nano Materials

The quality of the nano-suspension significantly influences the modifying effects of nano additives. Because of the high surface area and strong surface attraction force among particles, nanomaterials display a tendency of agglomeration and flocculation which brings the difficulty of well-distributing nanoparticles and obtaining homogeneous suspension [52]. Based on the methods suggested in most of the literature [26,53,54], in this study, superplasticizer and ultrasonic dispersion processing were adopted for acquiring high-quality suspension of nanomaterials before the fabrication of samples. Initially, the

weighed surfactant was mixed with one-third of the total amount of water in a beaker, followed by the addition of NS and CNT powder of different dosages. The solution was evenly and thoroughly stirred until the powder was dissolved and then the suspension was treated with ultrasonic energy. So as to avoid the impacts of heating on the solution, a UP50H ultrasonic apparatus with 50 W power and 30 kHz was used to ultrasonicate the solution for 30 min.

### 2.4. Fabrication of Samples

After the dispersion of nanomaterials, the remaining water and the nano-suspension were mixed into a container and stirred thoroughly for 2 min, followed by the addition of dry OPC and FAC. Then the mixture was homogeneously stirred with an electronic mixer for 3 min. Note that before adding into water and nanosuspension, the dry mix of cement and FAC were uniformly stirred for 2 min. Afterwards, the mixed cement mortar was poured into fully greased metal prism moulds of the size of 160 mm × 40 mm × 40 mm and laid on a flat surface. The moulds were filled with cement mixture in three layers for the purpose of compaction and subjected to a vibration table for approximately 2–3 min to evenly distribute the mixture. All moulds were then covered by plastic films to avoid moisture loss from evaporation and placed under room temperature for at least 24 h before demolding. After demolding, the casted samples were labelled and immediately put underwater in a curing tank for 28 days at room temperature in laboratory settings. The average hardened density of the mortar was 1092 kg/m$^3$ which can be classified as lightweight cement material [4]. Figure 2 shows the samples before the implementation of strength tests.

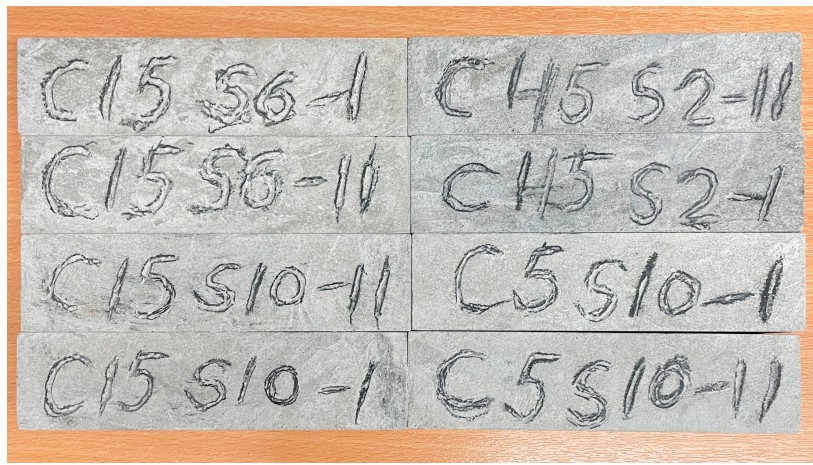

**Figure 2.** Specimens after 28 days.

### 2.5. Testing Methods

The test of flexural strength was implemented on a WDW-20 universal testing apparatus. The load application was performed at the rate of 2 mm per minute. For each mix design, five prisms were examined and the average values were documented as the flexural strength for each mix. Afterwards, six specimens were continuously subjected to compressive tests on the 50-C56G2 strength testing machine from CONTROLS. The strengths of the samples for each mix design were recorded, and the average value was used as the compressive strength. Four specimens were experimented on to obtain the water-absorption property of LWCM. Before immersion into the water, specimens were initially dried in the oven at room temperature and the weight of the samples was recorded every 24 h until the mass change was less than 1%. The samples were then submerged into water with the weight documented every 24 h. The tests were completed only after the 24 h mass change was less than 1%.

After the water absorption test, 4 representative specimens (C15S2, C5S0, RS, C45S10) were selected for implementing SEM, TG, and XRD tests. The specimens were first dried

in the oven at a temperature of 40 °C for 24 h. A small piece of the specimens was then acquired with a hammer and observed using a TM 3000 tabletop microscope from Hitachi to obtain the morphology of the samples. For TG and XRD tests, samples were ground into fine powder and passed through a 63 μm sieve in order to improve the consistency of the component proportion and ensure the accuracy of the test. TG examination was performed via Mettler/Toledo TG 3+ apparatus under ambient temperature rising from room temperature to 1000 °C with a heating rate of 10 °C/min. XRD was conducted on a Rigaku Ultima+ X-ray diffractometer with copper cathode radiation between diffraction angles (2θ) from 5° to 80°.

## 3. Results and Discussion

The outcomes obtained in the experiments are presented and discussed in this section, including flexural strength, compressive strength, and water absorption capacity. Microstructure and hydration mechanism are further analyzed via SEM, TG, and XRD tests.

### 3.1. Flexural Strength

The flexural strength of each mix is presented in Figure 3 and the changes in flexural performance due to the addition of nanomaterials are recorded in percentages as is shown in Figure 4. Figure 5 exhibits four representative samples after the flexural test. After failure, the surfaces of specimens C5S0 and C15S2 displayed comparatively less roughness than C45S10 and RS which were also of high spalling. This addressed the lower strength values obtained in the experiment for samples C45S10 and RS. The original data of the flexural strength test are recorded in Table A1, together with the standard deviation and weighting factors. T-test values are presented in Table A2. It can be clearly seen from Figures 3 and 4 that positive results were obtained in the experiments for the modified specimens, proving that the presence of nanomaterials effectively ameliorated the flexural capacity of LWCM with FAC. This is attributed to the outstanding characteristics of CNTs and NS which can greatly improve the hydration reaction and microstructure of LWCM. For sample C15S2, the maximal flexural strength of 2.13 MPa was measured, with an 89.02% increase compared to the control mix (1.13 MPa), while specimen C45S10 showed the lowest enhancement of flexural strength, with only 7.53% of growth to 1.21 MPa. Other dosages of nano additives contributed to the strength development of LWCM within the range of 10% to 50%.

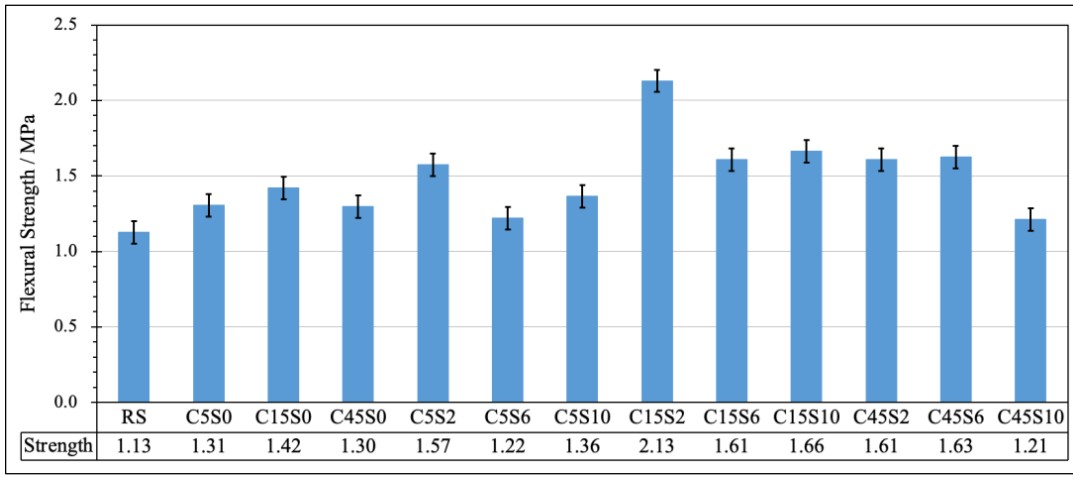

**Figure 3.** Flexural strength of LWCM.

On average, based on the acquired outcomes, specimens with the addition of 0.15% CNTs outperformed those with 0.05% or 0.45% CNTs, which suggests that 0.15% CNTs in this experiment is the optimal dosage in the hybrid use for improving flexural strength. Samples with constant 0.0%, 0.2%, and 1.0% of all NS exhibited an uptrend on flexural strength by 8.85%, 35.37%, and 21.87%, respectively, followed by a decline by 0.12, 0.52, and 0.45 MPa,

accordingly, when the dosage of CNTs grew from 0.05% to 0.15% to 0.45%. Specimens with 0.6% of NS presented an increasing tendency initially and then maintained the same level. The initial improvement in strength can be ascribed to the positive effect of CNTs to bridge the microcracks in cement materials and introduce additional closing against crack propagation [55,56]. Further, the inferior flexural behaviors of samples with 0.45% CNTs are associated with exceeding the amount of CNTs which can agglomerate and flocculate, negatively affecting the hydration process and further hindering the strength gain [57]. Furthermore, the sliding of CNTs in the cement matrix under growing loads results in a weak bond between CNTs and the matrix, negatively influencing the strength of NS-modified cement materials [58].

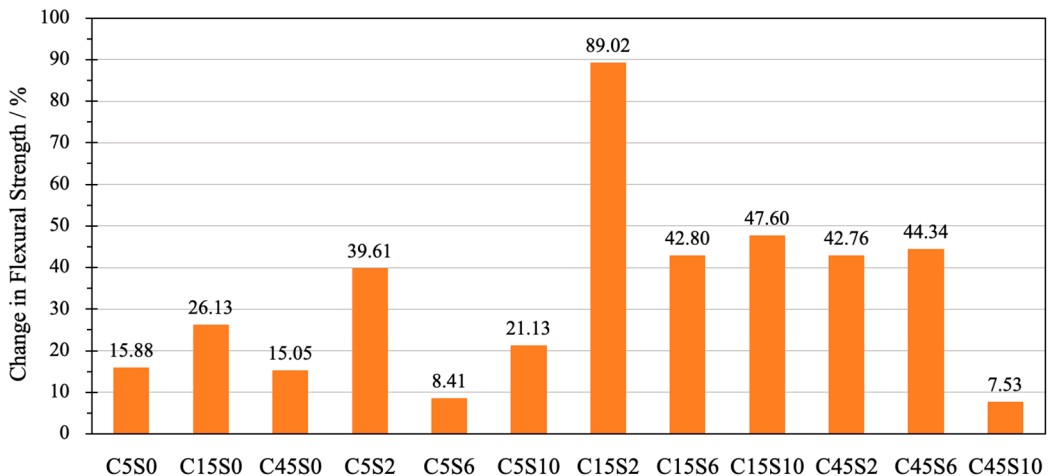

**Figure 4.** Change in flexural strength of LWCM.

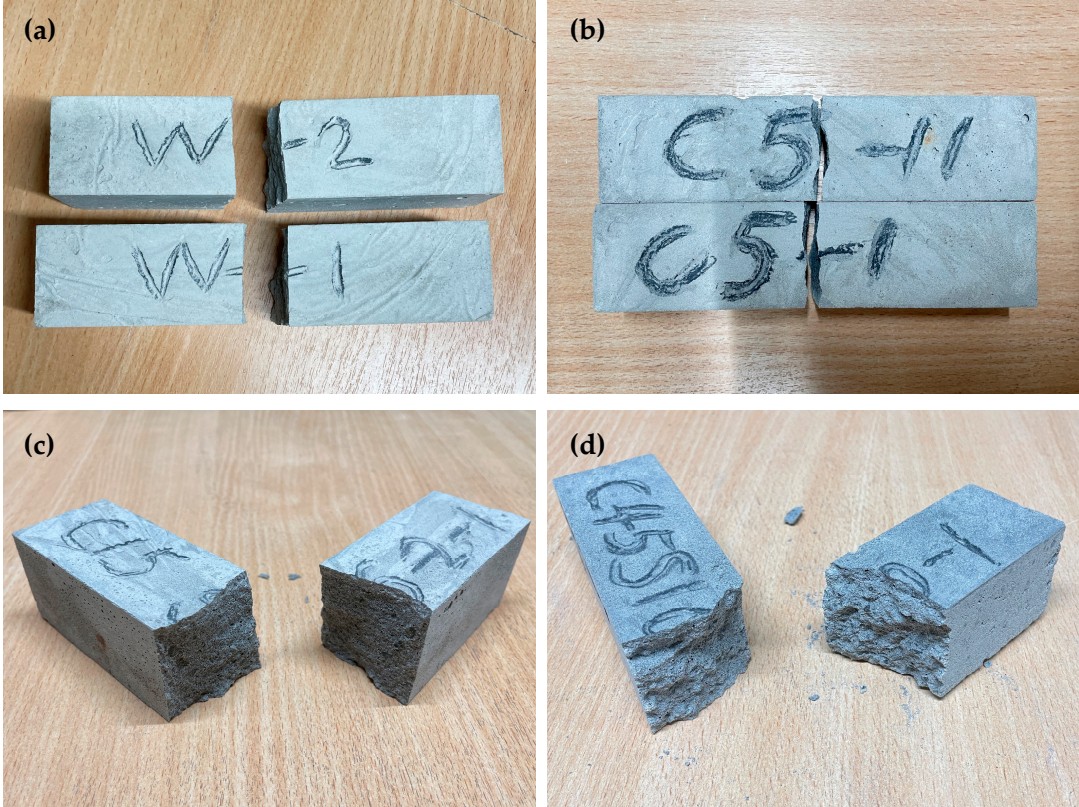

**Figure 5.** Degradation of representative specimens after the flexural test: (**a**) RS; (**b**) C5S0; (**c**) C15S2; (**d**) C45S10.

The dosage of 0.2% NS bcw shows a moderately better modifying impact than other dosages, which highlights that 0.2% content of NS in the hybrid additives is optimal for the enhancement of flexural strength. For samples containing constant 0.15% CNTS, the incorporation of NS in three dosages all accounted for the further enhancement of flexural strength, with the most impressive increase, 49.95%, measured for the sample C15S2. Reductions of strength were only observed for samples C5S6 and C45S10, in contrast with the specimens without NS. This shows that the use of NS cannot always lead to further improvement of flexural performance in binary usage together with CNTs. The initial strength improvement of adding 0.2% NS is related to the ameliorated adhesion between cement matrix and CNTs, but with the growing amount of NS added to LWCM, the tendency of NS to absorb water leads to an incomplete hydration reaction, negatively affecting strength [24,59,60].

### 3.2. Compressive Strength

The compressive strength of LWCM is recorded in Figure 6 for each mix, and the percentage changes in strength are exhibited in Figure 7. The original data from the compressive strength test are presented in Table A3, along with the values of standard deviation and weighting factors of each data group. T-test values were calculated in Table A4. Figure 8 displays the crack patterns of specimens RS, C5S0, C15S2, and C45S10 after compressive strength tests. It can be seen that compared to the other three specimens, sample C45S10 showed the greatest degradation after loading, corresponding to its lowest strength value.

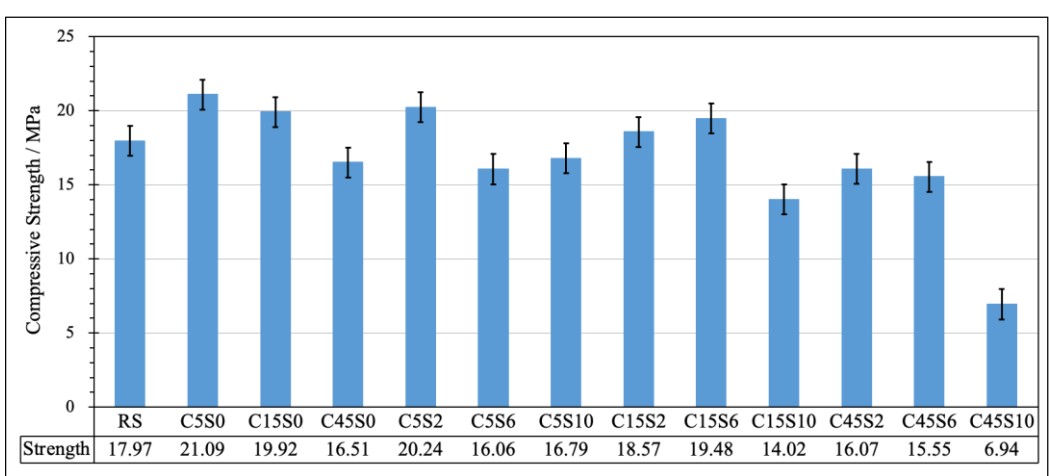

**Figure 6.** Compressive strength of LWCM.

As can be seen in Figures 6 and 7, the hybrid use of nano additives cannot guarantee the improvement of compressive strength, and certain dosages of CNTs and NS can be associated with negative influence on strength gain. Five mixes (C5S0, C15S0, C5S2, C15S2, C15S6) demonstrated enhancement in compressive strength while the other seven (C45S0, C5S6, C5S10, C15S10, C45S2, C45S6, C45S10) indicated reduction. It is worth noticing that all specimens containing 0.45% of CNTs presented a decline in strength, which can be ascribed to the fact that an exceedingly high amount of CNTs leads to the agglomeration and increasing difficulty of dispersion, which causes agglomeration in the composites [59,61]. Further, the incorporation of 1.0% NS resulted in different extents of strength loss regardless of the dosage of CNTs. Similarly, this can be attributed to the high amount of NS that adds up to the difficulty of dispersion due to agglomeration, and accounts for a weak zone, leading to the initiation of microcracks and the decline of strength [50]. The most considerable enhancement in compressive strength was measured for the specimen containing only 0.05% of CNTs and no NS, with a strength increase of 17.35% to 21.09 MPa compared to that of the reference sample (17.97 MPa); while by adding

0.45% CNTs and 1.0% NS, the greatest decrease in strength was observed, with the value declining by 61.36% to 6.94 MPa. Other dosages of nano additives were in correspondence with the growth of compressive strength in a range of 1% to 10% as well as a strength reduction by 10% to 20%.

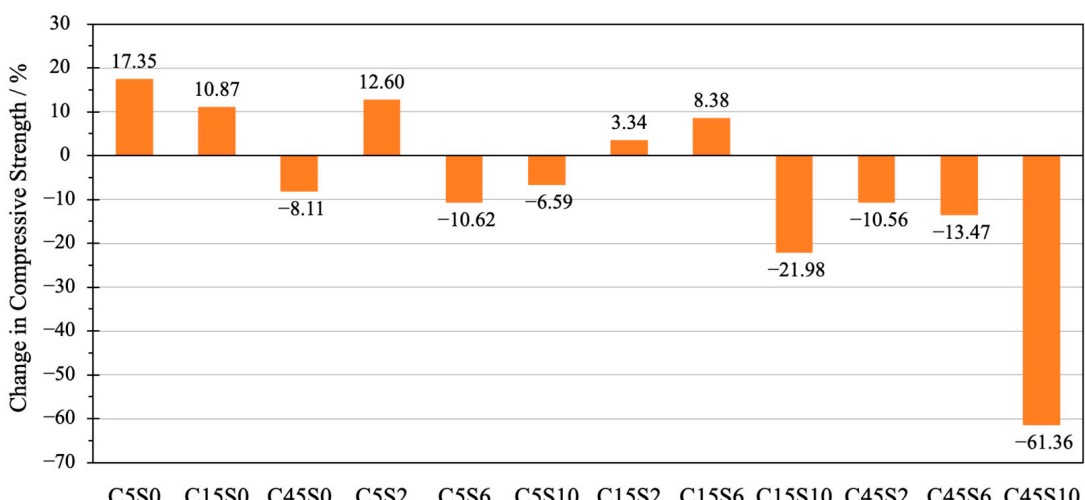

**Figure 7.** Changes in compressive strength of LWCM.

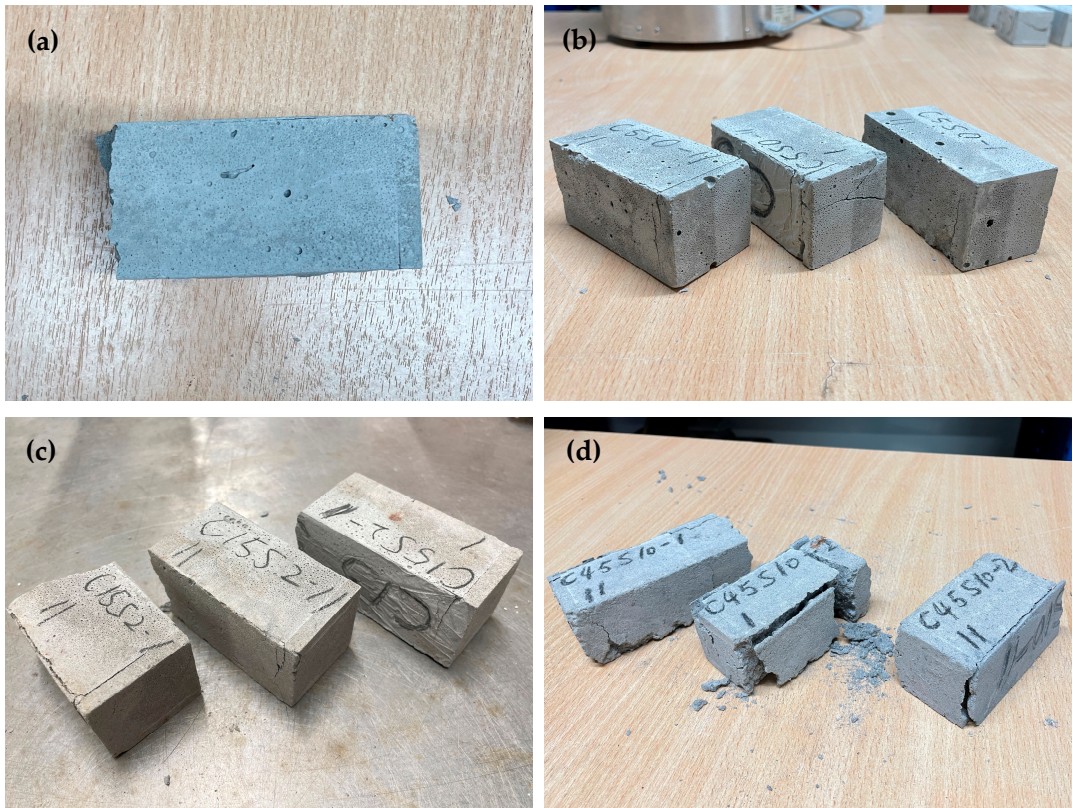

**Figure 8.** Degradation of representative specimens after compressive test: (**a**) RS; (**b**) C5S0; (**c**) C15S2; (**d**) C45S10.

At the constant amounts of NS of 0.2% and 1.0%, the growth of CNT content from 0.05% to 0.45% contributed to the decrease of compressive strength by 20.60% and 58.67%, respectively. For specimens containing a constant content of 0.6% NS, nevertheless, the compressive strength of LWCM showed an increase to 19.48 MPa when the dosage of CNTs

grew from 0.05% to 0.15%. Afterwards, a 20.17% reduction in strength was observed as the content of CNTs increased to 0.45%. These findings indicate that despite the increasing amount, in the presence of NS, the synergic use of CNTs can partially account for the decline of compressive strength. Karakouzian et al. [37] ascribed this to the existing Van der Waals force that hinders the homogeneous dispersion of CNTs, negatively affecting the development of the strength.

Compared to the samples with only CNTs, the addition of NS at all contents in the presence of CNTs led to the loss of compressive strength. This states that the hybrid incorporation of both CNTs and NS has inferior effects on the improvement of compressive behaviors compared to the single utilization of CNTs. According to Karakouzian et al. [37], the high demand for water by NS in cement composites can lead to an adverse influence on mechanical properties, and such a negative impact can also affect the extent of enhancement due to binary addition of CNTs and NS. For a constant amount of CNTs equal to 0.05%, along with the growth of NS content to 0.6% bcw, the compressive strength of LWCM declined by 23.85% to the lowest measured value of 16.06 MPa, in contrast with that containing no NS (21.09 MPa), after which a slight improvement in strength by 4.54% was observed. For specimens with a constant content of 0.15% and 0.45% of CNTs, a similar evolution was illustrated, with initially a mild fluctuation when the amount of NS was increased to 0.6%. Then a sharp reduction in strength was observed (28.03% and 55.37%, respectively) at the NS dosage of 1.0%. Apart from the nature of NS to absorb water, the decrease in strength is also ascribed to the poorly dispersed NS particles at the high amount as well as the lack of CH to further react with the unconsumed NS, negatively affecting the compressive properties [62].

*3.3. Water Absorption*

Water absorption of concrete indicates the durability of cement materials and is correlated to pore structure as well as the microvoids existing in the hardened concrete [63,64]. The original data from the water absorption tests are recorded in Appendix A (Tables A5–A7). In Figure 9, the characteristics of water absorption of LWCM are assessed via the weight of absorbed water, as well as the percentage of absorbed water, which is the percentage ratio of the weight difference to the dry weight [38]. The weight difference is the value between the sample weight after submerging into the water and the weight after drying up. As can be seen in the bar graph, in contrast with the referencing specimen, with 17.2 g of water absorbed in the test, the addition of nanomaterials contributed to the decline of water absorbed by the LWCM on different levels. The specimen C45S10 exhibited the lowest decrease in water absorption, of approximately 1.2 g, showing that at high dosage, the synergic incorporation of CNTs and NS failed to reduce the number of microvoids in the cement mortar. This also addresses the high weight of absorbed water (11.5 g) in the sample including only 0.45% of CNTs. Another reason can be ascribed to the nature of nanomaterials that require a high amount of water. Specimen C5S10 shows the lowest percentage of water absorption, 4.3%, demonstrating that the synergic presence of nano additives at a proper amount is effective for improving the pore structure of cement mortar and decreasing porosity. No obvious trend was observed between the amount of NS or CNTs and the water absorption capacity in the sample. Only when the CNTs content is constant at 0.05% and 0.15% were average greater synergic effects measured.

Figure 10 illustrates the change in the percentage water absorbed in comparison with the reference sample. On average, specimens with 0.05% and 0.15% of CNTs displayed comparatively higher loss in percentage water absorption, with the most remarkable value observed for samples C5S2 and C5S10, with 11.40% and 11.52%, respectively. This can be associated with the increasingly dense microstructure due to the incorporation of nanofillers at a proper dosage. The changing values varied within the range from 7% to 12%, except specimens C45S0 and C45S10, which exhibited a high percentage of water absorption, with only 5.21% and 0.30% of loss, respectively. This can be related to the loose morphology of the two specimens, as is proven in the SEM images.

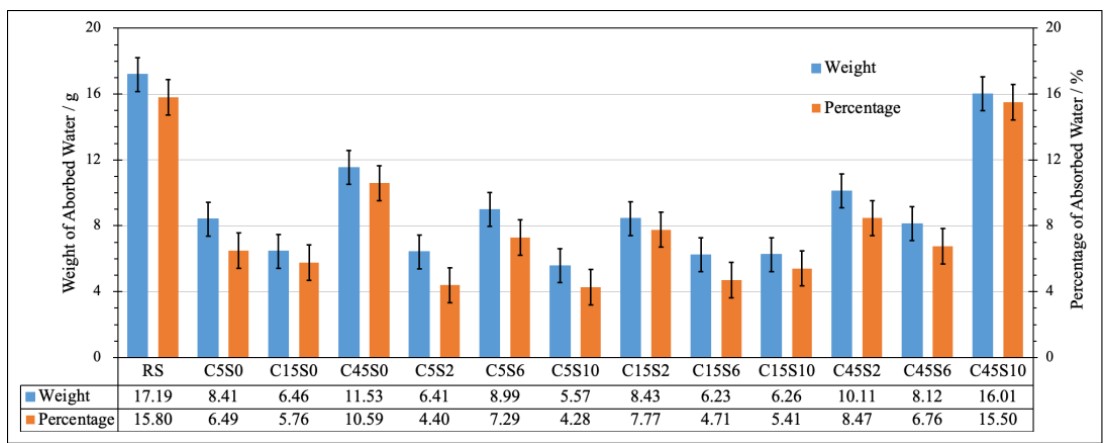

**Figure 9.** Results of water absorption test.

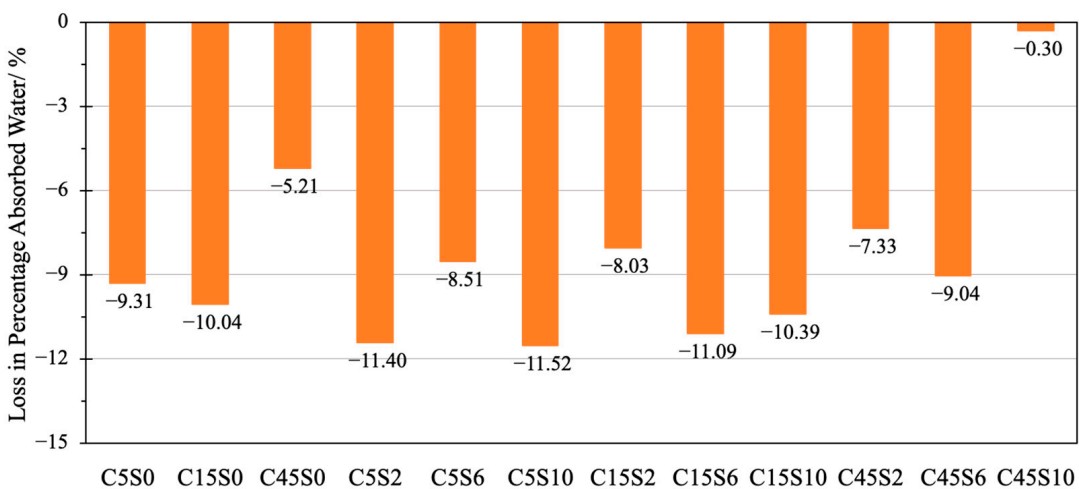

**Figure 10.** Change in percentage of absorbed water.

Figure 11 exhibits the amount of water absorbed by the samples along with the time. As can be seen in the graph, for all the samples, within 24 h, the same sharp increment of the amount of water absorbed was observed, followed by a slight enhancement in the upcoming days. This indicates that the samples have reached their saturation to absorb water at the early stage. After that, the capacity of water absorption of samples is limited. Other than the reference sample, regardless of the NS content, specimens containing 0.45% of CNTs all exhibited greater absorbing velocity compared to others with 0.05% and 0.15% CNTs and absorbed a higher amount of water at saturation. This indicates that 0.45% dosage of CNTs is less effective to improve the porosity of LWCM in contrast with other CNT contents.

*3.4. Thermalgravimetric Analysis (TG)*

Thermalgravimetric analysis was carried out on specimens RS, C5S0, C15S2, and C45S10 to evaluate the characteristics of substances and the material transformations during the hydration process. The aim of the test was to reveal the synergic impacts of CNTs and NS on the cement hydration reaction, especially their effects as nucleation sites. According to Yao and Lu [54], as well as Jung et al. [65], the main reasons resulting in the loss of sample weight during the increment process of temperature are the evaporation of free water in the cement matrix as well as the dehydration of bound water from hydration products and their decomposition. Figure 12 illustrates the thermogravimetric (TG) and differential thermogravimetric (DTG) curves acquired in the tests from room temperature

to 1000 °C. Along with the increase of ambient temperature, three characteristic peaks in each curve appear at the temperature range from 50 to 230 °C, 400 to 500 °C, and 630 to 780 °C. The occurrence of the representative peaks of weight loss indicates the important transformation of hydration products. The mass change within the temperature scope between 50 to 230 °C is related to the volatilization of free water from the LWCM [54], as well as the decomposition of hydrates, especially CSH [65–67]. The second peak in the range of temperature rising from 400 to 500 °C can be characterized by the dehydration of hydration product CH from LWCM [54,68], and the peak with temperature varying from 630 to 780 °C is ascribed to the decomposition phase of calcium carbonate ($CaCO_3$) [54,67].

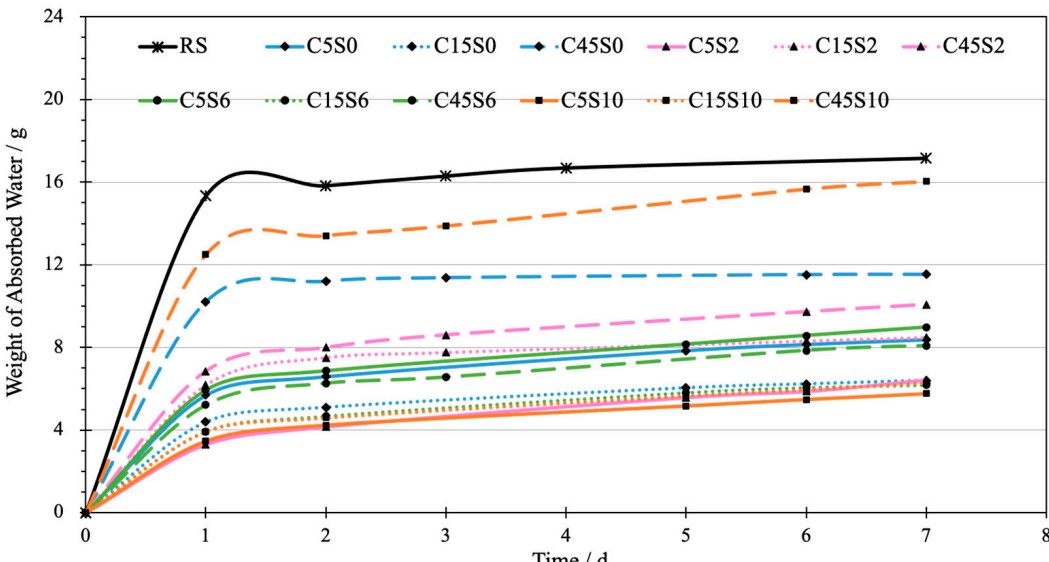

**Figure 11.** Evolution of absorbed water weight with the change of time.

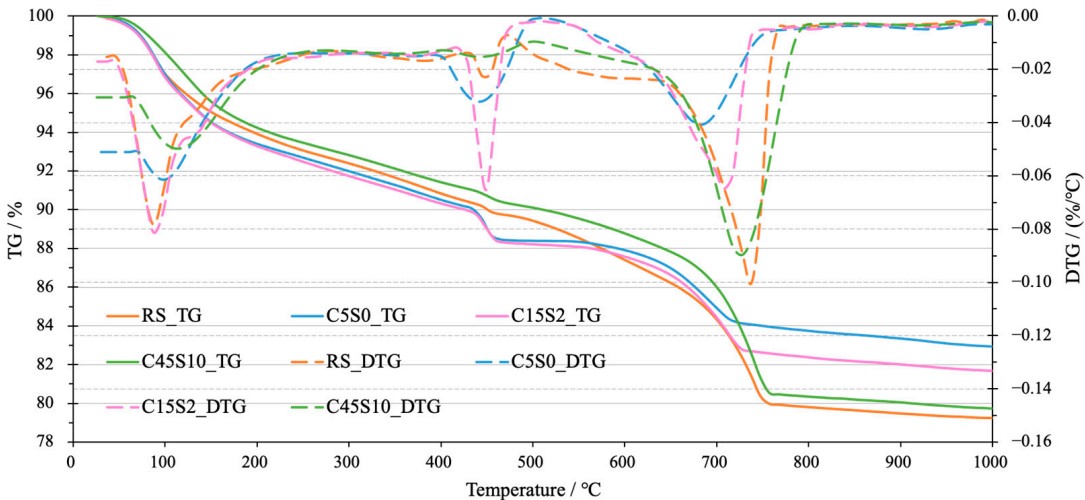

**Figure 12.** Results from the thermogravimetric analysis.

According to Carriço et al. [66] and Alarcon-Ruiz et al. [68], the loss of weight in the initial two temperature extents (100–230 °C and 400–500 °C) can approximately represent the degree of hydrate dehydration (Hdh) and dihydroxylation (Hdx) during the cement hydration process, respectively. In Figure 13, the values of Hdh and Hdx were calculated and presented. Compared to the control specimen (3.05% Hdh and 0.90% Hdx), samples C5S0 and C15S2 exhibited both higher dehydration and dihydroxylation values, with increments of 22.57% and 29.51% for Hdh, as well as 128.18% and 113.97% for Hdx. This

confirms the existing higher hydration degree in LWCM due to the incorporation of CNTs and CNT-NS additives, resulting in an increasing amount of hydration products. This is associated with the effects of CNTs and NS acting as nucleation sites and activators, providing an auxiliary environment and promoting the process of hydration as well as the formation of hydrates. The greatest Hdh value, 3.95% was measured for sample C15S2, which demonstrates that the presence of NS led to developed synergic influence and further improved cement hydration, resulting in a higher amount of CSH generated. Sample C5S0 showed the most remarkable Hdx value of 2.06%, indicating the decreased amount of CH in the hydration with the inclusion of NS. This can be related to the silica pozzolanic reaction of NS which consumes CH and produces CSH during the hydration process [59]. Another point worth noticing is that for sample C45S10, the Hdh value showed the least increment compared to the other two modified samples, and the Hdx value was even lower than that of the control specimen, showing the limited improvement in hydration degree and fewer generated hydrates. This decline of CSH and CH can be ascribed to the great water demand for nanomaterials that compete with cement for water during the hydration process. Further, the agglomeration of CNTs and NS at high dosage is another thing that can hinder cement hydration.

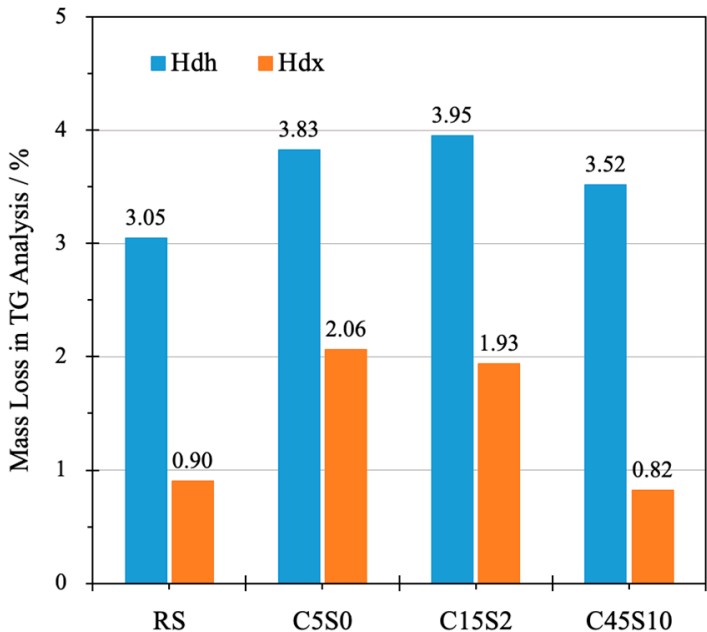

**Figure 13.** Dehydration and dihydroxylation values in TG analysis.

*3.5. Scanning Electron Microscope (SEM)*

In Figure 14, the Micrographs of specimens RS, C5S0, C45S10, and C15S2 at the magnification of 250, 500, and 1000 were scanned via scanning electron microscope (SEM) to evaluate the dispersion quality of nanomaterials as well as their interaction with the cementitious matrix.

Three morphologies of FAC are observed: intact spherical FAC, broken FAC owing to the hollow shell structure that tends to be crushed under loading, and partially reacted FAC. According to Hanif et al. [69], FAC displays some extent of pozzolanic activity, attributed to the presence of amorphous silica, as well as lime, and can participate in the hydration reactions to produce CH and CSH. The use of FAC led to loose microstructure and highly porous structure, especially the agglomeration of FAC accounted for the propagation of large voids in the cement mortar, and the fraction of FAC can increase the porosity of the system. Due to the glassy surface and physicomechanical characteristics of FAC, the bond between aggregates and cement matrix tend to be loose and microcracks take place in the mortar.

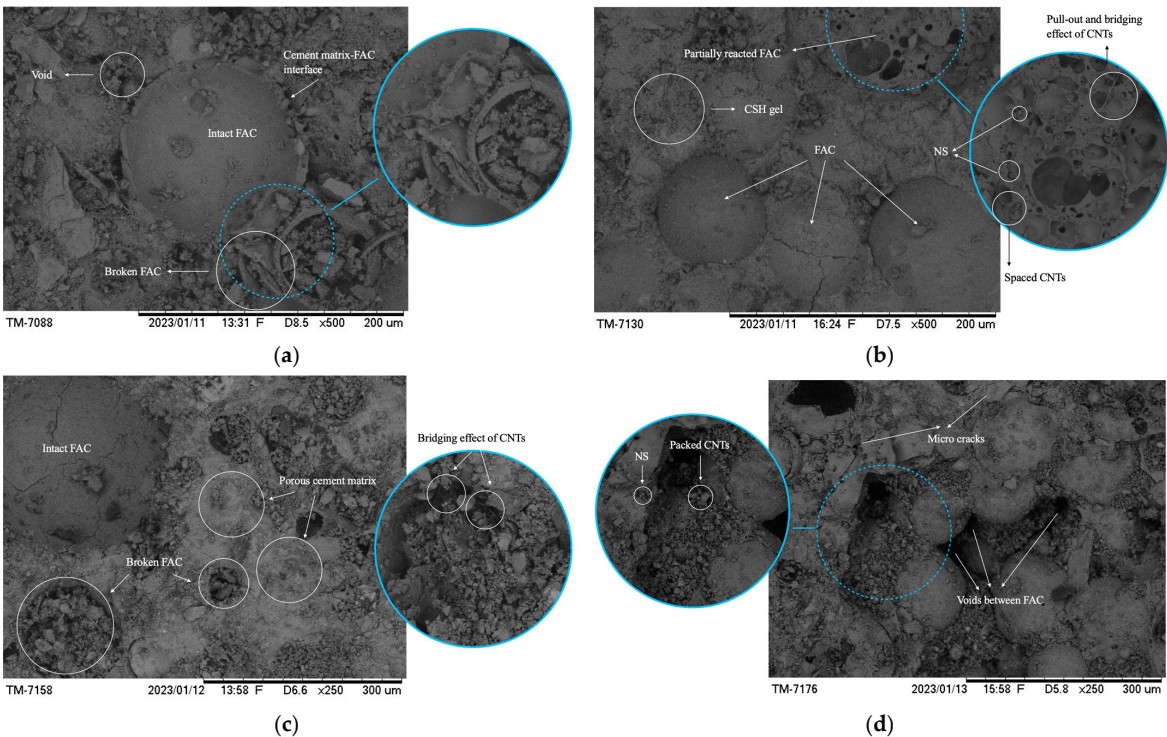

**Figure 14.** Morphology of LWCM: (**a**) RS; (**b**) C15S2; (**c**) C5S0; (**d**) C45S10.

In contrast with the reference sample (Figure 14a), with the addition of NS and CNTs, the micro-cracks and voids in sample C15S2 (Figure 14b) were reduced. NS showed the filler effects, filling up the gaps as well as spaces between the cementitious matrix and FAC, leading to a denser microstructure and increasing the bond strength. Because of the impacts of NS to accelerate the hydration process, more homogeneous and compact CSH pastes were formed in LWCM, decreasing the porosity and improving the pore structure. This further contributed to the enhancement in compressive strength. The bridging and pull-out effects of CNTs were seen in the SEM graphs which effectively bridged the gaps and restrained the propagation of micro-cracks, increasing the bond strength of cementitious matrix and that between matrix and lightweight aggregates. Spaced CNTs filled the voids of cement matrix with an influence similar to fibres, impressively enhancing the flexural performance of LWCM. Similar effects of CNTs were observed for sample C5S0 in Figure 14c which showed a dense and homogeneously distributed porous cement matrix, greatly contributing to the growth of strength.

In specimen C45S10 (Figure 14d) a growing amount of large voids and wide micro-cracks were observed compared to the control sample, resulting in an increasingly porous and uneven microstructure. This is in correspondence with the high water absorption value obtained in the tests. This can be ascribed to the exceeding amount of nano additives in LWCM which compete with cementitious materials for the water, adversely hindering the process of hydration reaction, and contributing to fewer hydration products. The effects of NS performing as nanofillers were witnessed but failed to considerably improve the microstructure. Additionally, packed CNTs were seen in the images, indicating that at high dosages, CNTs fail to evenly disperse in the cement matrix and form agglomeration, which results in less effective bridging impacts and negatively affects the microstructure of LWCM.

### 3.6. X-ray Diffraction Analysis (XRD)

In Figure 15, the XRD patterns of the representative specimens RS, C5S0, C45S10, and C15S2 are presented. According to Liew et al. [70] as well as Yao and Lu [54], the improvement of the mechanical performance of modified cement-based materials is affected

by both the physical interaction between nano additives and cementitious matrix, and also the chemical transformation during the hydration reaction. The objective of the experiment was to reveal the influence of NS and CNTs on the transformation of chemical compositions owing to the hydration process. The chemical identification of the peaks was carried out via XRD analysis software HighScore Plus 3.0 and relevant literature sources [54,67,71–73].

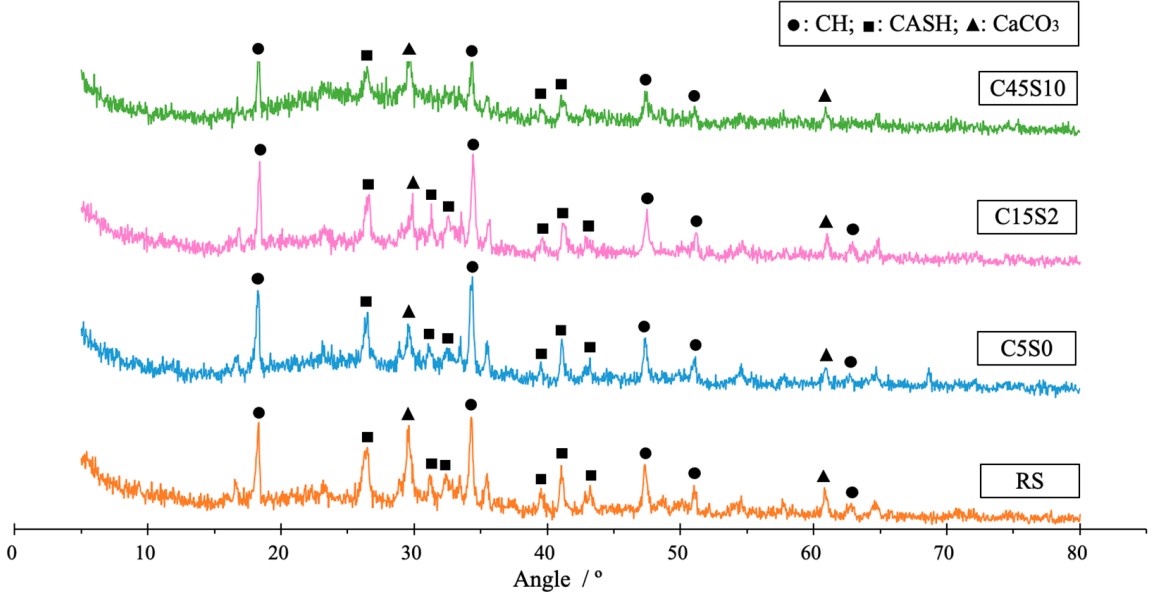

**Figure 15.** XRD patterns of LWCM.

Based on the curves in Figure 15, three characteristic peaks of minerals were identified, indicating the formation of CH, calcium aluminum silicate hydrate (CASH), and $CaCO_3$ in LWCM. In contrast with the pattern of the reference sample, the intensity of CH and CASH peaks exhibited a slight increase for specimen C5S0 and C15S2, but no remarkable change was observed, and no occurrence of new peaks was identified. This indicates that although the presence of CNTs and NS can facilitate cement hydration, the degree is limited. Considering the evolution of compressive and flexural curves, it can be concluded that the effect of physical interaction due to CNTs and NS that fill the voids and bridge the cracks is greater in the cement matrix than that in chemical transformation. However, when the content of CNTs and NS was increased to 0.45% and 1.0%, respectively, for sample C45S10, peaks of hydration products CASH at around 31°, 33°, and 43°, as well as CH at approximately 63°, disappeared on the pattern. Further, the intensity of the remaining peaks that belong to CH and CASH all displayed different extents of decrease, showing the decline of the hydration reaction and formation of hydration products. This is in accordance with the results obtained in TG analysis and testifies to the adverse effects of excessive addition of CNTs and NS which hinders cement hydration. The examination of XRD only determines the existence of crystal phases, notwithstanding, according to Karpova [67], at the 2θ degree in the range from 28.7 to 35.0, the amorphous background in XRD patterns can demonstrate the presence of amorphous CSH. From the patterns acquired in XRD tests, it can be seen that a large amount of amorphous CSH was yielded during the hydration process for all four samples, and, qualitatively, specimens C15S2 and C5S0 were linked to greater intensity of amorphous CSH compared to that of RS and C45S10.

## 4. Conclusions

In this experimental study, a sustainable green LWCM was developed and can be used for further production of LWC. The engineering application of the cement mortar includes, but is not limited to, cold-bridge materials, the manufacture of cement blocks for exterior and interior finishing, and structural materials for building components that

require comparatively less load-bearing capacity, such as lintels of windows and doors. The cement mortar was designed to be prefabricated in factory conditions; nevertheless, onsite production can be also achieved with the preparation of nanosuspension in advance. Based on the obtained results, primary conclusions can be drawn as follows:

- In contrast with the reference sample, specimens with nano additives presented development in flexural strength at various extents, but the binary effects of CNTs and NS cannot always guarantee greater flexural behaviour than that of specimens with only CNTs. The most remarkable flexural strength was measured for specimen C15S2, with an 89.02% increment observed, whereas specimen C45S10 exhibited the lowest improvement in flexural strength with only an increase of 7.53%;

- In comparison to the control specimen, enhancement and reduction in compressive strength were both measured. Regardless of the NS content, the presence of 0.45% CNTs always accounted for a decline of strength, owing to the increasing difficulty of dispersion and agglomeration of CNTs. Similarly, despite the changes in CNT content, samples with 1.0% NS presented decreases in compressive strength. The most impressive improvement was measured for the sample C5S0, with a growth of 17.35% to 21.09 MPa;

- Compared to the compressive strength of samples with only CNTs, the synergic addition of NS cannot further improve the strength. The greatest value of LWCM under the synergic use of CNTs and NS was measured for specimen C5S2, with the value increasing by 12.60%. Overall, the binary usage of NS and CNTs was more effective to improve flexural behaviour than compression. This can be attributed to the bridging effect of CNTs, which is more dominant than the filling and nucleation effects of CNTs and NS on the enhancement of mechanical characteristics;

- The incorporation of nano additives accounted for the reduction of absorbed water of modified specimens. Sample C5S2 showed the greatest decrease in the percentage of absorbed water, viz., 11.40%. This can be related to the increasingly dense and homogeneous microstructure. Sample C45S10 showed the lowest reduction of absorbed water, with only a decrease of 0.30%. For all the samples, the capacity of absorbing water reached saturation at an early stage within 24 h, after which the increment of water absorbed was very limited;

- Three morphologies of FAC were observed in the micrographs via microscope: intact spherical FAC, broken FAC, and partially reacted FAC. The existence of FAC led to the loose and highly porous microstructure. The agglomeration of FAC promoted the propagation of large voids and cracks. The filler effects of NS and CNTs were observed for the modified samples, which effectively filled up the voids in LWCM, forming a denser microstructure. The bridging and pull-out effects of CNTs were identified in the micrographs, which effectively restrained the propagation of microcracks, benefiting the development of flexural strength.

- TG analysis demonstrated that CNTs can effectively improve the hydration degree in LWCM, and the synergic use of NS further enhanced this positive effect, owing to the nucleation effects of nano additives. Sample C15S2 showed the highest amount of CSH formed during hydration process, and specimen C5S0 had the greatest amount of CH. This is associated with the influence of NS which participates in the reaction, consuming CH and producing CSH.

- XRD analysis proved the positive effects of CNTs and NS on promoting cement hydration, but the degree was limited. Combined with the results from TG and SEM, it can be inferred that the physical interaction of CNTs and NS in the cement matrix is greater than the chemical reaction.

- Based on this study, it is suggested to further experiment at a smaller content gradient of CNTs varying from 0.05% to 0.15%, and NS around 0.2% bcw, to identify the optimal content for synergic use. The usage of FAC and nano additives in cement materials are regarded as sustainable and economical. Further attempts can be made to compare environmental impacts and economic viability.

**Author Contributions:** Conceptualization, A.K., Y.D.; methodology, A.K., Y.D.; software, Y.D.; validation, A.K., Y.D.; formal analysis, A.K., Y.D.; investigation, A.K., Y.D.; resources, A.K.; data curation, A.K., Y.D.; writing—original draft preparation, Y.D.; writing—review and editing, A.K.; visualization, Y.D.; supervision, A.K.; project administration, A.K.; funding acquisition, A.K. All authors have read and agreed to the published version of the manuscript.

**Funding:** This research was funded by the Department of Building Materials and Products, Institute of Materials and Structures, Faculty of Civil Engineering, Riga Technical University.

**Institutional Review Board Statement:** Not applicable.

**Informed Consent Statement:** Not applicable.

**Data Availability Statement:** Main experimental data have been included in Appendix A.

**Acknowledgments:** The authors wish to express their sincere appreciation to all the staff in the Department of Building Materials and Products for their assistance.

**Conflicts of Interest:** The authors declare no conflict of interest.

## Appendix A

In Appendix A, the original data acquired from the strength tests and water absorption experiments are supplemented. As is shown in Table A1, the original tested flexural strengths of five sample groups are recorded, together with the average strength values, the corresponding standard deviations, and the weighting factors of each data group. Table A2 displays the T-test results of flexural strength. Five data groups contributed to the same significance to determine the average value of the strength; thus, the weighting factor of the data from each group was equal to 0.2. Tables A3 and A4 include the original data acquired from compressive strength tests, as well as the relevant statistical analysis. In Table A3, six groups of data, the average strength values, and the standard deviations are displayed. The weighting factor of each data group was the same 0.17, for each group was of equal importance. Table A4 illustrates the values of the T-test. It can be seen that the difference in data between any two groups was small. The original data from the water absorption tests were recorded in Tables A5 and A6. The T-test values are calculated in Table A7.

**Table A1.** Original data of flexural strength test (unit: MPa).

| Sample | Flexural Strength | | | | | Average Strength | Standard Deviation |
|---|---|---|---|---|---|---|---|
| | **A** | **B** | **C** | **D** | **E** | | |
| RS | 1.127 | 0.792 * | 1.140 | 1.115 | 1.124 | 1.13 | 0.01 |
| C5S0 | 1.304 | 1.307 | 1.301 | 1.310 | 1.306 | 1.31 | 0.01 |
| C15S0 | 1.425 | 1.448 | 1.463 | 1.371 | 1.399 | 1.42 | 0.04 |
| C45S0 | 1.298 | 1.286 | 1.341 | 1.249 | 1.307 | 1.30 | 0.03 |
| C5S2 | 1.566 | 1.586 | 1.560 | 1.578 | 1.574 | 1.57 | 0.01 |
| C5S6 | 1.194 | 1.247 | 1.184 | 1.275 | 1.207 | 1.22 | 0.04 |
| C5S10 | 1.385 | 1.347 | 1.249 | 1.463 | 1.379 | 1.36 | 0.08 |
| C15S2 | 2.130 | 2.099 | 2.072 | 2.198 | 2.147 | 2.13 | 0.05 |
| C15S6 | 1.587 | 1.610 | 1.590 | 1.609 | 1.647 | 1.61 | 0.02 |
| C15S10 | 1.640 | 1.647 | 1.659 | 1.662 | 1.706 | 1.66 | 0.03 |
| C45S2 | 1.249 * | 1.524 | 1.563 | 1.636 | 1.710 | 1.61 | 0.08 |
| C45S6 | 1.310 * | 1.654 | 1.561 | 1.684 | 1.605 | 1.63 | 0.05 |
| C45S10 | 1.179 | 1.261 | 1.251 | 1.156 | 1.209 | 1.21 | 0.05 |
| Weighting factor /% | 20 | 20 | 20 | 20 | 20 | - | - |

*: the data were excluded in statistical analysis.

**Table A2.** T-test results of flexural strength test.

| T-Test | A, B | A, C | A, D | A, E | B, C | B, D | B, E | C, D | C, E | D, E |
|--------|------|------|------|------|------|------|------|------|------|------|
| Value | 0.97 | 0.94 | 0.90 | 0.92 | 0.86 | 0.89 | 0.89 | 0.79 | 0.78 | 0.99 |

**Table A3.** Original data of compressive strength test (unit: MPa).

| Sample | Compressive Strength | | | | | | Average Strength | Standard Deviation |
|--------|------|------|------|------|------|------|------------------|--------------------|
| | A | B | C | D | E | F | | |
| RS | 18.14 | 19.15 * | 17.79 | 17.86 | 17.99 | 18.07 | 17.97 | 0.14 |
| C5S0 | 21.38 | 20.88 | 21.04 | 21.13 | 20.95 | 21.15 | 21.09 | 0.18 |
| C15S0 | 19.88 | 20.02 | 19.87 | 19.93 | 19.96 | 19.88 | 19.92 | 0.06 |
| C45S0 | 16.41 | 16.53 | 16.68 | 16.49 | 16.37 | 16.60 | 16.51 | 0.12 |
| C5S2 | 20.06 | 20.48 | 20.06 | 20.25 | 20.37 | 20.19 | 20.24 | 0.17 |
| C5S6 | 15.86 | 16.10 | 16.24 | 16.09 | 15.91 | 16.17 | 16.06 | 0.15 |
| C5S10 | 16.82 | 16.68 | 16.86 | 16.90 | 16.71 | 16.74 | 16.79 | 0.09 |
| C15S2 | 18.62 | 18.70 | 18.43 | 18.51 | 18.64 | 18.52 | 18.57 | 0.10 |
| C15S6 | 19.08 | 19.80 | 19.22 | 19.58 | 19.41 | 19.76 | 19.48 | 0.29 |
| C15S10 | 14.00 | 14.03 | 14.02 | 14.06 | 14.01 | 14.00 | 14.02 | 0.02 |
| C45S2 | 15.88 | 16.30 | 16.23 | 16.13 | 15.92 | 15.97 | 16.07 | 0.17 |
| C45S6 | 14.52 * | 15.48 | 15.61 | 15.57 | 15.53 | 15.56 | 15.55 | 0.05 |
| C45S10 | 6.98 | 6.88 | 7.91 * | 7.01 | 6.85 | 7.00 | 6.94 | 0.14 |
| Weighting factor/% | 17 | 17 | 17 | 17 | 17 | 17 | - | - |

*: the data were excluded in statistical analysis.

**Table A4.** T-test results of compressive strength test.

| T-Test | A, B | A, C | A, D | A, E | A, F | B, C | B, D | B, E | B, F | C, D | C, E | C, F | D, E | D, F | E, F |
|--------|------|------|------|------|------|------|------|------|------|------|------|------|------|------|------|
| Value | 0.88 | 0.98 | 0.94 | 0.99 | 0.93 | 0.95 | 0.99 | 0.95 | 0.99 | 0.97 | 0.99 | 0.96 | 0.96 | 1.00 | 0.96 |

**Table A5.** Original data of absorbed water weight in water absorption test (unit: g).

| Sample | Weight of Absorbed Water | | | | Average Weight | Standard Deviation |
|--------|------|------|------|------|----------------|--------------------|
| | A | B | C | D | | |
| RS | 17.16 | 17.35 | 17.02 | 17.21 | 17.19 | 0.14 |
| C5S0 | 8.38 | 8.71 | 8.53 | 8.02 | 8.41 | 0.29 |
| C15S0 | 6.40 | 6.09 | 6.81 | 6.52 | 6.46 | 0.30 |
| C45S0 | 11.53 | 11.89 | 11.02 | 11.68 | 11.53 | 0.37 |
| C5S2 | 6.42 | 6.91 | 6.32 | 6.00 | 6.41 | 0.38 |
| C5S6 | 8.97 | 9.21 | 8.75 | 9.03 | 8.99 | 0.19 |
| C5S10 | 5.76 | 6.12 | 5.28 | 5.11 | 5.57 | 0.46 |
| C15S2 | 8.47 | 8.59 | 8.26 | 8.41 | 8.43 | 0.14 |
| C15S6 | 6.17 | 5.77 | 6.45 | 6.53 | 6.23 | 0.34 |
| C15S10 | 6.28 | 5.66 | 6.34 | 6.75 | 6.26 | 0.45 |
| C45S2 | 10.07 | 10.16 | 9.81 | 10.39 | 10.11 | 0.24 |
| C45S6 | 8.08 | 8.67 | 7.52 | 8.21 | 8.12 | 0.47 |
| C45S10 | 16.03 | 16.05 | 15.82 | 16.14 | 16.01 | 0.14 |
| Weighting factor/% | 25 | 25 | 25 | 25 | - | - |

**Table A6.** Original data of percentage absorbed water in water absorption test (unit: %).

| Sample | Percentage of Absorbed Water | | | | Average Weight | Standard Deviation |
|---|---|---|---|---|---|---|
| | A | B | C | D | | |
| RS | 15.61 | 15.58 | 16.43 | 15.57 | 15.80 | 0.42 |
| C5S0 | 6.47 | 6.49 | 6.48 | 6.53 | 6.49 | 0.02 |
| C15S0 | 5.80 | 6.09 | 4.69 | 6.46 | 5.76 | 0.76 |
| C45S0 | 10.65 | 10.34 | 11.12 | 10.26 | 10.59 | 0.39 |
| C5S2 | 4.14 | 4.07 | 4.60 | 4.77 | 4.40 | 0.34 |
| C5S6 | 7.42 | 7.05 | 7.49 | 7.20 | 7.29 | 0.20 |
| C5S10 | 4.36 | 4.13 | 4.16 | 4.45 | 4.28 | 0.16 |
| C15S2 | 7.83 | 7.43 | 8.15 | 7.66 | 7.77 | 0.30 |
| C15S6 | 4.83 | 4.95 | 4.63 | 4.42 | 4.71 | 0.24 |
| C15S10 | 5.56 | 5.73 | 5.24 | 5.09 | 5.41 | 0.29 |
| C45S2 | 8.54 | 8.01 | 9.07 | 8.27 | 8.47 | 0.45 |
| C45S6 | 6.78 | 6.32 | 7.60 | 6.34 | 6.76 | 0.60 |
| C45S10 | 15.59 | 15.27 | 15.99 | 15.17 | 15.50 | 0.37 |
| Weighting factor/% | 25 | 25 | 25 | 25 | - | - |

**Table A7.** T-test results of water absorption test.

| T-Test | A, B | A, C | A, D | B, C | B, D | C, D |
|---|---|---|---|---|---|---|
| Weight of absorbed water | 0.94 | 0.92 | 0.99 | 0.86 | 0.95 | 0.91 |
| Percentage of absorbed water | 0.91 | 0.92 | 0.94 | 0.84 | 0.97 | 0.86 |

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
