# Peer review of "Synergic Effects of Nano Additives on Mechanical Performance and Microstructure of Lightweight Cement Mortar"

_applsci, doi:10.3390/app13085130_

Round 1
Reviewer 1 Report
The authors have carried out an experimental study to analyze the synergic effect of nano additives on the mechanical and microstructure performance of lightweight concrete. Different mechanical and microstructure tests were conducted, including flexural strength, compressive strength, water absorption, SEM, TGA, and XRD. The authors are expected to address the below-given comments, provide justifications to the raised points, and make references to the various ideas and suggestions as proposed to improve the quality and readability of this manuscript before it can be considered for re-evaluation. The comments of this reviewer are as under:
1) What specific types of carbon nanotubes (CNTs) were used in this study, and how were they synthesized and characterized?
2) Can you describe the process by which the pore structure of the concrete is improved by the inclusion of CNTs and NS? How does this contribute to the improvement of mechanical properties?
3) What are the potential limitations or drawbacks of using industrial waste fly ash cenosphere as a lightweight aggregate in concrete?
4) What are the potential limitations or drawbacks of using industrial waste fly ash cenosphere as a lightweight aggregate in concrete?
5) Line 32-33: provide the reference, i.e., https://doi.org/10.1016/j.conbuildmat.2021.126061.
6) Statistical analysis is essential to ensure the repeatability of tests, particularly flexural and compressive strength tests. Publishing the current research requires presenting model summaries, weighting factors, parameter estimates, and t-tests.
7) Corresponding to different mix designs, the C15S2 outcomes the peak flexural strength, whereas in compressive strength results, the same superior behavior of C15S2 is not reflected. Can authors explain why this behavior is and what is the reason behind it?
8) The XRD results show a higher peak of Portlandite which shows that the secondary hydration reaction is still undergoing. Whereas there is no peak pertaining to CSH/CASH gel marked, which is majorly responsible for mechanical strength development. Authors should give further consideration/analyze the XRD results again to further verify the mineralogy.
9) There are many grammatical and structural errors. Pay attention!
Author Response
Dear reviewer!
Thank you for your comments and questions. We decided to revise the name of the manuscript. The replies and explanations are in the cover letter.
Best regards
Aleksandrs

Reviewer 2 Report
C1) As a first comment, the authors stress that a lightweight concrete is under development and study, but there is no information whether the hardened mixture fall in that category. There are numerous standards on concrete underlying the corresponding range for density for a such concrete.
C2) It is not mentioned very clear what type of lightweight concrete is involved. There are some ways of getting such concretes, the classic one using natural or artificial light aggregate or replacing the river aggregate in some percentages with some artificial light or ultralight materials, for example polystyrene (EPS).
C3) In the title is used the word "concrete". According to classical theory, I mention here Adam Neville's book, if there is no coarse aggregate, then the material can be denominated mortar. In this study the coarse aggregate is not used, therefore, is the meaning concrete appropriate used by the authors in this manuscript ?
C4) The authors must be very sure about meaning of the word "synergic". Some explaining sites gives the following for synergic, I quote "involving the interaction or cooperation of two or more organizations, substances, or other agents to produce a combined effect greater than the sum of their separate effects:". I presume that the authors agree. I presume that the title should be related mainly to the obtained results not to the expectation before the study. Therefore if the results do not have the declared synergic characteristic, than is obvious that the relevance of the word in the title is very low.
C5) Reading the Introduction, it is quite cumbersome to catch the research gap and why is necessary to study a such topic. Involving nanomaterials which are quite costly it should be done to get something significant in terms of the mechanical or physical properties, but at a glace in Fig. 4, it can be seen that for example the compressive strengths is quite poor. Considering the information presented in Fig. 4, the grade of concrete cannot fall even into the structural grade. Working with such low concrete strengths the importance of the study, if any, is low.
C6) There are no photos on the appearance of this material nether before nor after failure of the test specimens. The authors should have thought that a such material must be well illustrated and presented.
C7) There is nothing mentioned about a the civil engineering applications of this material. If there are, are they important? Why it should be used a quite complicated obtaining material with a such low compressive strength involving nanomaterials without real possibilities to decrease w/c ratio because otherwise a well dispersion of the nanomaterial is not assured, for example CNT.
C8) A quite serious lack in presentation of the results is the missing of the tables with values in figures regarding the investigation properties as the flexural and compressive strength, and the water absorption. If the manuscript will be accepted, then the authors of some potential future studies will be unable to make accurate comparisons with the results which are illustrated as charts only.
C9) The information presented in Table 3 is far to be clear. In order to avoid confusion, if the w/c ratio is mentioned, then the cementing material is only OPC otherwise the water/binder ratio must be mentioned and the binder content must be written down. Also the information presented in FAC's columns is confusing. The authors should indicate what is the aggregate's column?
C10) Is it enough a such cement content per cubic meter? Is it hardly to believe that a cement content of 270 kg/mc assure well durability over the time and other performances. Perhaps the authors consider that a part of the FAC is also cementing material, therefore the previous comment has some reason. If 270 is not the cement content per cubic meter, then the authors should have mentioned it.
C11) The phrase from line 173 to 174 is confusing.
C12) The information presented at the heading 2.3 to 2.4 shows that manufacture process is a cumbersome one. For a building material which can be manufactured many times on-site it seems to be a cumbersome process. Perhaps the material has some particular applications, it is not intended to be one of large use, but the authors must underline this.
C13) The results regarding the strength properties are weird. First, for the reference mixture, the compressive to flexural strength ratio is quite small, 2.44. It is hardly to believe that for a quite low compressive strength, a such high tensile strength by flexure is obtained. For the rest of the mixtures, from C5S0 to C45S6 the mentioned ratio decreases from 2.72 to 1.44. The most weird result is the last one where the tensile strength by flexure is higher than the compressive strength.
C14) The compressive strength is obtained testing some prism ends instead of standard test specimens, cubes or cylinders.
C15) The investigation seems to be wrong designed in order to catch the synergic effect. If this would be done, then the effect of each nanomaterial apart must be studied followed by the both's effect.
Considering that there are serious flaws in the results and considering that the authors made a study of which importance is ambiguous, my first recommendation is rejecting the study in this shape.
Author Response
Thank you for your comments and questions. We decided to revise the name of the manuscript. The replies and explanations are in the cover letter.
Best regards,
Aleksandrs

Round 2
Reviewer 1 Report
The authors have addressed my comments. The paper is acceptable for publication in its current form.
Reviewer 2 Report
Thanks to the corresponding author for the given answers. The corresponding author responded explicitly to all raised issues. The title and content of the article was modified accordingly in a satisfactory shape.